# Predator-secreted sulfolipids induce defensive responses in *C. elegans*

Zheng Liu[1], Maro J. Kariya[2], Christopher D. Chute[3], Amy K. Pribadi[1,4], Sarah G. Leinwand[1], Ada Tong[1], Kevin P. Curran[1], Neelanjan Bose[2], Frank C. Schroeder[2], Jagan Srinivasan[3] & Sreekanth H. Chalasani[1,4]

Animals respond to predators by altering their behavior and physiological states, but the underlying signaling mechanisms are poorly understood. Using the interactions between *Caenorhabditis elegans* and its predator, *Pristionchus pacificus*, we show that neuronal perception by *C. elegans* of a predator-specific molecular signature induces instantaneous escape behavior and a prolonged reduction in oviposition. Chemical analysis revealed this predator-specific signature to consist of a class of sulfolipids, produced by a biochemical pathway required for developing predacious behavior and specifically induced by starvation. These sulfolipids are detected by four pairs of *C. elegans* amphid sensory neurons that act redundantly and recruit cyclic nucleotide-gated (CNG) or transient receptor potential (TRP) channels to drive both escape and reduced oviposition. Functional homology of the delineated signaling pathways and abolishment of predator-evoked *C. elegans* responses by the anti-anxiety drug sertraline suggests a likely conserved or convergent strategy for managing predator threats.

[1] Molecular Neurobiology Laboratory, The Salk Institute for Biological Studies, La Jolla, CA 92037, USA. [2] Boyce Thompson Institute and Department of Chemistry and Chemical Biology, Cornell University, Ithaca, NY 14853, USA. [3] Department of Biology and Biotechnology, Worcester Polytechnic Institute, Worcester, MA 01605, USA. [4] Division of Biological Sciences, University of California, San Diego, La Jolla, CA 92093, USA. These authors contributed equally: Zheng Liu, Maro J. Kariya, Christopher D. Chute. Correspondence and requests for materials should be addressed to S.H.C. (email: schalasani@salk.edu)

Animal survival depends on the ability to sense predators and generate appropriate behavioral and physiological changes[1]. Such defensive behaviors[2], including the commonly observed 'flight or freezing' responses, are often hardwired into the genome of the prey—for example, mice reliably exhibit fear-like responses to cat odors despite not having encountered cats for hundreds of generations[3]. Despite this, the neuronal and signaling machinery that regulate defensive behaviors remains poorly understood. One approach to uncovering the nature of innate defensive responses is to identify the molecular signals between predators and prey and map the underlying neuronal and molecular machinery that drive defensive responses to these signals.

Studies from both vertebrates and invertebrates indicate that signaling between predators and prey involves multiple sensory modalities including vision, audition, and most frequently olfaction[4–6]. Considerable progress has been made in identifying the sensory neurons that detect predator-released odors in several model systems. For example, in mice, the chemosensory neurons in the vomeronasal organ (VNO), Grueneberg ganglion, and main olfactory epithelium have been shown to facilitate defensive behaviors through detection of signals from cat urine and fox feces[3,7,8]. These neurons project to higher brain regions where predator odor information is processed to generate stereotyped defensive behaviors[9]. More generally, it is thought that circuit constancy typically accompanies behavioral stereotypy. While neural circuits that detect odors vary between individuals[10], those sensing predator-released odors appear to be invariant between members of the same species[3]. However, the precise identities of the participating neurons, their connections, and the nature of the circuit computations driving these invariant defensive behaviors have remained elusive.

We approached these questions by analyzing the behavioral responses of the nematode Caenorhabditis elegans[11] to a predatory nematode Pristionchus pacificus[12]. These two nematodes likely shared a common ancestor around 350 million years ago[13]. Recent studies have shown that P. pacificus is a facultative predator. P. pacificus can bite and kill C. elegans, a process facilitated by the extensive re-wiring of the P. pacificus nervous system under crowded and/or starvation conditions[14,15]. C. elegans, with its fully mapped neural network comprising of just 302 neurons connected by identified synapses and powerful genetic tools, is ideally suited for a molecular and circuit-level analysis of complex behaviors[16,17]. Combining chemical and genetic methods, we dissected the signaling circuits underlying C. elegans' responses to P. pacificus. We found that a novel class of sulfated small molecules excreted by P. pacificus trigger defensive responses in C. elegans. These P. pacificus-derived chemical signals are detected by C. elegans via multiple sensory neurons and processed via conserved signaling pathways.

## Results

**A predator elicits defense responses in C. elegans.** C. elegans was originally isolated from compost heaps in the developmentally arrested dauer stage[18]. However, recent studies have isolated proliferating and feeding populations of C. elegans from rotting flowers and fruits[19], where they are often found to cohabit with other nematodes including the Diplogastrid Pristionchus (M-A. Felix, personal communication). Previous reports have shown that the terrestrial nematode, P. pacificus can kill and consume the smaller nematode C. elegans[20]. We hypothesized that the prey, C. elegans, detects the predator, P. pacificus, through chemical cues and thus tested C. elegans responses to P. pacificus excretions. We found that C. elegans showed immediate avoidance upon perceiving excretions of starved, but not well-fed

predators (Fig. 1a and Supplementary Fig. S1a). Excretions from P. pacificus collected after 21 h of starvation ('predator cue') were found to consistently repel genetically diverse C. elegans isolates (Supplementary Fig. S1b). Next, we tested whether volatile components could be responsible for the activity of the predator cue by analyzing prey responses using a chemotaxis assay optimized for volatiles. We found that predator cue had no significant effect on C. elegans taxis responses in this assay (Supplementary Fig. S1c,d), indicating that volatiles do not contribute to the activity of predator cue. Together, these results show that starving P. pacificus release potent non-volatile repellent(s) that induce rapid C. elegans avoidance.

We further found that C. elegans exposed to predator cue did not lay eggs for many minutes following exposure, even when placed on food (bacterial lawn), suggesting that predator cue-induced stress affects egg-laying behavior. Consistent with this idea, previous studies have shown that C. elegans retain eggs in the gonad when exposed to environmental stressors[21]. To test our hypothesis, we designed a behavioral assay wherein the prey was exposed to predator cue for 30 min, and egg-laying was monitored for many hours following cue removal. Animals exposed to predator cue laid significantly fewer eggs than controls during the initial 60 min following cue removal. During the next hour (i.e., the 60–120 min post-cue time period), these animals laid more eggs than controls, suggesting that predator cue transiently modified egg-laying behavior, but not egg production (Fig. 1b). Collectively, these results indicate that starving P. pacificus release a potent, non-volatile factor (predator cue) that elicits multiple prey responses, namely urgent escape behavior followed by up to one hour of reduced egg laying.

**Predator-derived sulfolipids elicit defense responses.** We aimed to identify the chemical structure(s) of the small molecule(s) excreted by P. pacificus that cause C. elegans avoidance behavior. As the P. pacificus exo-metabolome is highly complex, consisting of more than 20,000 distinct compounds detectable by UHPLC-HRMS (Fig. 1c), we used a multistage activity-guided fractionation scheme (see Supplementary Methods). After three rounds of fractionation, chemical complexity of individual fractions had been reduced sufficiently to enable comparative metabolomics analysis of 2D NMR spectra[22,23] and high-resolution tandem mass spectrometry data of active and adjacent inactive fractions (Fig. 1d, e, Supplementary Tables S1–S3). This analysis revealed several novel (ω-1)-branched-chain sulfolipids (sufac#1, and sufal#2) as major components of active, but not inactive fractions (Fig. 1f). Further analysis revealed several additional sulfolipids with closely related structures, including sufac#2 and sufal#1, which differ in the position of attachment of the sulfate moiety (Fig. 1h). Next, we corroborated the structures of these compounds via total synthesis and tested their activity in the avoidance assay. The terminal alcohols sufal#1 and sufal#2 accounted for most of the isolated activity (Supplementary Fig. S3a, Supplementary methods).

None of the identified P. pacificus sulfolipids could be detected in the metabolomes of Escherichia coli OP50 (used as food for nematodes) (Supplementary Fig. S2), C. elegans, or several other nematode species (Supplementary Table S4), which were extracted and analyzed under identical conditions. Notably, the identified sulfolipids are structurally similar to sodium dodecyl sulfate (SDS, Fig. 1i), which is a potent C. elegans avoidance cue[24].

**Sulfolipids are perceived by redundant sensory neurons.** To define the prey neural circuit that detects predator cue, we tested the role of all 12 pairs of amphid sensory neurons, which project dendrites to the nose of the animal to sense environmental

changes (Fig. 2a)[16,25]. Previous studies have shown that sensory neurons in the amphid ganglia located in the head of the worm detect repellents and generate reversals in an attempt to avoid the noxious cues[26]. We generated animals missing each of the 12

amphid neuron pairs and tested their ability to respond to predator cue. We found that animals lacking pairs of ASJ, ASH, ASI, or ADL neurons were defective in their responses to predator cue (Fig. 2b, c), indicating that *C. elegans* uses multiple sensory

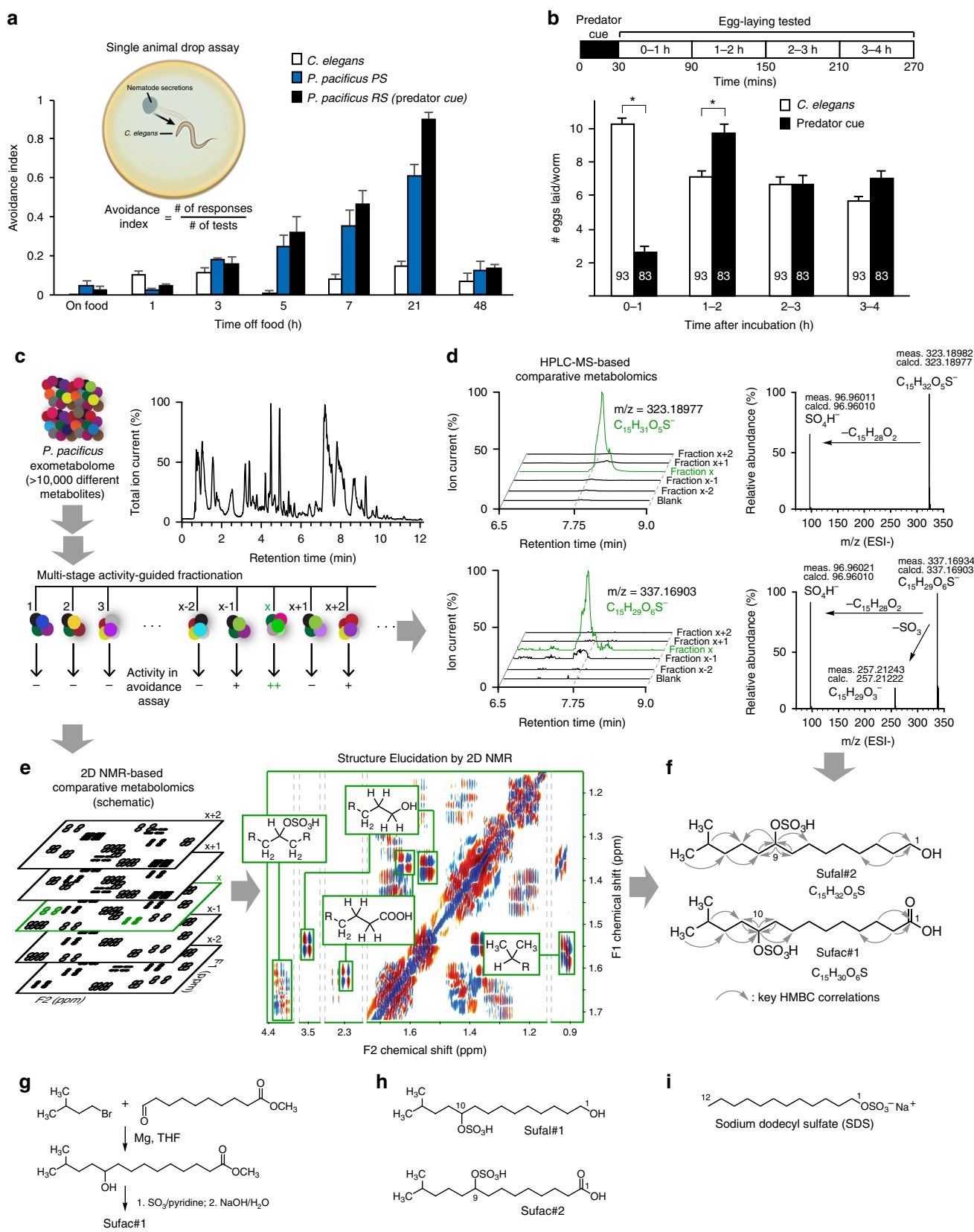

neurons to detect predators. Responses to sulfolipids purified from predator cue and SDS were similarly reduced in animals lacking these neurons (Supplementary Fig. S3f, g). In contrast, animals missing any of the other 8 neuronal pairs showed normal responses suggesting that these neurons were not required for avoidance to predator cue.

To confirm the involvement of the ADL, ASH, ASI, and ASJ neurons, we monitored their responses to predator cue using calcium imaging[27]. Calcium responses are strongly correlated with neuronal activity in *C. elegans* neurons[28]. We found that adding predator cue to the nose of the prey activated ADL and ASH (Fig. 2d, e, Supplementary Fig. S4 for all traces), whereas predator cue removal activated ASI and ASJ neurons (Fig. 2f, g, Supplementary Fig. S4 for all traces). Also, whereas ADL and ASJ responded to both tested dilutions of predator cue (Fig. 2d, g), ASH and ASI only detected the more concentrated cue (i.e., 1:10 dilution, but not 1:50) (Fig. 2e, f), suggesting different response thresholds for these four neuronal pairs. Collectively, these results show that addition of predator cue activates ADL and ASH neurons, whereas its removal increases ASI and ASJ activity.

**_C. elegans_ defensive responses require CNG and TRP channels.** To gain insight into the signal transduction machinery underlying these responses, we examined the behavior of mutants lacking specific signaling components. We found that mutants lacking the alpha subunit (*tax-4*), but not the beta subunit (*tax-2*), of the cyclic nucleotide-gated (CNG) ion channel exhibited defective responses to predator cue (Fig. 3a). Previously, TAX-4 but not TAX-2 subunits have been shown to form a homomeric CNG channel[29], suggesting that TAX-4 might function in a TAX-2 independent manner. Moreover, expressing the full-length *tax-4* cDNA via a *tax-4* promoter, an ASI-specific promoter, or an ASJ-specific promoter (but not via an ASH-selective promoter) restored normal behavior to the null mutants (Fig. 3a). The ability of these transgenes to rescue avoidance behavior was largely dose dependent, as it varied depending on the amount of *tax-4* transgene expressed in ASI and ASJ neurons (Supplementary Fig. S5a). These data indicate that increased CNG signaling from ASI neurons could compensate for the lack of signaling from ASJ, and vice-versa. Collectively, these results show that TAX-4 functions in a TAX-2 independent manner in ASI and ASJ neurons to drive predator avoidance. Similarly, mutants lacking the transient receptor potential (TRP) channel OCR-2, but not OSM-9, were defective in their responses to predator cue. Further, we observed that OCR-2 functions in ADL and ASH neurons, but not in ASI or ASJ neurons (Fig. 3b), and that responses to ASH- and ADL-specific *ocr-2* transgenes were also dose dependent (Supplementary Fig. S5b), indicating that signaling from ASH could compensate for the lack of ADL signaling, and vice-versa. Testing samples of purified sulfolipids confirmed that *tax-4* (but

not *tax-2*) and *ocr-2* (but not *osm-9*) mutants were defective in avoidance to these molecules (Supplementary Fig. S5c). Together, these results show that OCR-2 functions in a OSM-9 independent manner in ASH and ADL neurons to generate avoidance to predator cue, results consistent with previous studies[30,31].

To investigate possible interactions between CNG and TRP channel signaling, we analyzed *tax-4;ocr-2* double mutants. Restoring TAX-4 function to ASI neurons and OCR-2 to ASH neurons (in combination) using the highest dosage of the respective transgenes conferred normal predator cue avoidance to the double mutants (Fig. 3c). Moreover, partial avoidance was seen for other rescue combinations (TAX-4 in ASJ and OCR-2 in ASH; TAX-4 in ASI and OCR-2 in ADL, and TAX-4 in ASI alone) (Fig. 3c). Together, these data indicate that there are at least four neuronal signaling pathways that can drive robust avoidance to *Pristionchus* cue: (1) ASI sensory neurons using TAX-4 channels; (2) ASI and ASH neurons using TAX-4 and OCR-2 channels, respectively; (3) ASJ and ASH using TAX-4 and OCR-2 channels, respectively; and (4) ASI and ADL using TAX-4 and OCR-2 channels, respectively (Fig. 3d). Similarly, we found that *tax-4* mutants, but not *ocr-2* mutants, did not curtail their egg-laying behavior (a longer-lasting effect) in response to predator cue, and that restoring TAX-4 function to ASI or ASJ significantly improved this *tax-4* defect (Fig. 3e, f). These results show that signaling from a subset of the sensory circuitry that drives instantaneous avoidance (ASI or ASJ, but not ASH or ADL) is required to generate the long-lasting changes in egg-laying behavior.

**Sertraline acts on GABA signaling to block prey behavior.** To identify signaling pathways regulating responses to predator cue, we screened a library of human anti-anxiety drugs since these compounds have previously been shown to attenuate predator-induced defensives responses in a prey[32]. In this screen, wild-type animals were pre-treated with different compounds for 30 min before testing their responses to predator cue. In a pilot screen of 30 compounds (Supplementary Table S5), we found that pre-treating prey with a selective serotonin reuptake inhibitor (SSRI), sertraline (brand name 'Zoloft') attenuated avoidance to predator cue and purified sulfolipids (Fig. 4a, Supplementary Fig. S6a). Sertraline also attenuated avoidance responses to fructose, and, to a lesser extent, $CuSO_4$ (Fig. 4a), suggesting that this drug affects some, but not all repellent circuits. Suppression of avoidance behavior by sertraline was dependent on drug concentration (Supplementary Fig. S6b) and lasted at least 30 min after the drug was removed (Supplementary Fig. S6c). To test whether sertraline modifies signaling from specific sensory neurons, we analyzed mutants expressing different rescuing transgenes. Sertraline had no detectable effect on the behavior of *tax-4* or *ocr-2* mutants, but it attenuated avoidance to predator cue of *tax-4* mutants

**Fig. 1** Predator-released sulfolipids drive *C. elegans* behaviors. **a** *C. elegans* avoid excretions from starving *P. pacificus* PS (PS312, domesticated) and RS (RS5725B a wild isolate, predator cue) strains. Inset shows a schematic of the avoidance assay. **b** Top shows schematic of the modified egg-laying assay and bottom, *C. elegans* lay fewer eggs after a 30 min exposure to concentrated predator cue, but recovers after 2 h. **c** UHPLC-HRMS analysis reveals a complex mixture of >10,000 metabolites, which was subjected to multistage activity-guided fractionation using reverse-phase chromatography. After four fractionation steps, most of the activity (++) was found in fraction x. Averages and s.e.m. are shown. n > 30 for each condition. **d** UHPLC-HRMS ion chromatograms (*m/z* value ± 5 ppm) of active fraction x and adjacent fractions for two sulfate-containing metabolites that were strongly enriched in the active fraction (left). MS–MS analysis (right) confirms presence of sulfate moieties in both compounds. **e** Schematic representation of 2D NMR-based comparative metabolomics (left) of consecutive fractions ($x - 2$ to $x + 2$) used to identify signals specific to fraction x. Cropped 2D NMR (dqfCOSY) spectrum (right) of active fraction highlighting signals that represent specific features of the identified metabolites (gray lines define edges of shown subsections). **f** Chemical structures of metabolites identified via comparative metabolomics from active fraction x, sufac#1 and sufal#2. Gray arrows indicate important correlations observed in heteronuclear 2D NMR (HMBC) spectra. **g** Synthesis of sufac#1; THF: tetrahydrofuran. **h** Homologous metabolites sufac#2 and sufal#1 were also detected by UHPLC-HRMS. **i** Chemical structure of sodium dodecyl sulfate (SDS). Averages and s.e.m. are shown and number of animals tested are indicated on each bar or condition. *$p < 0.05$ obtained by comparison with controls using Fisher's exact t-test with Bonferroni correction

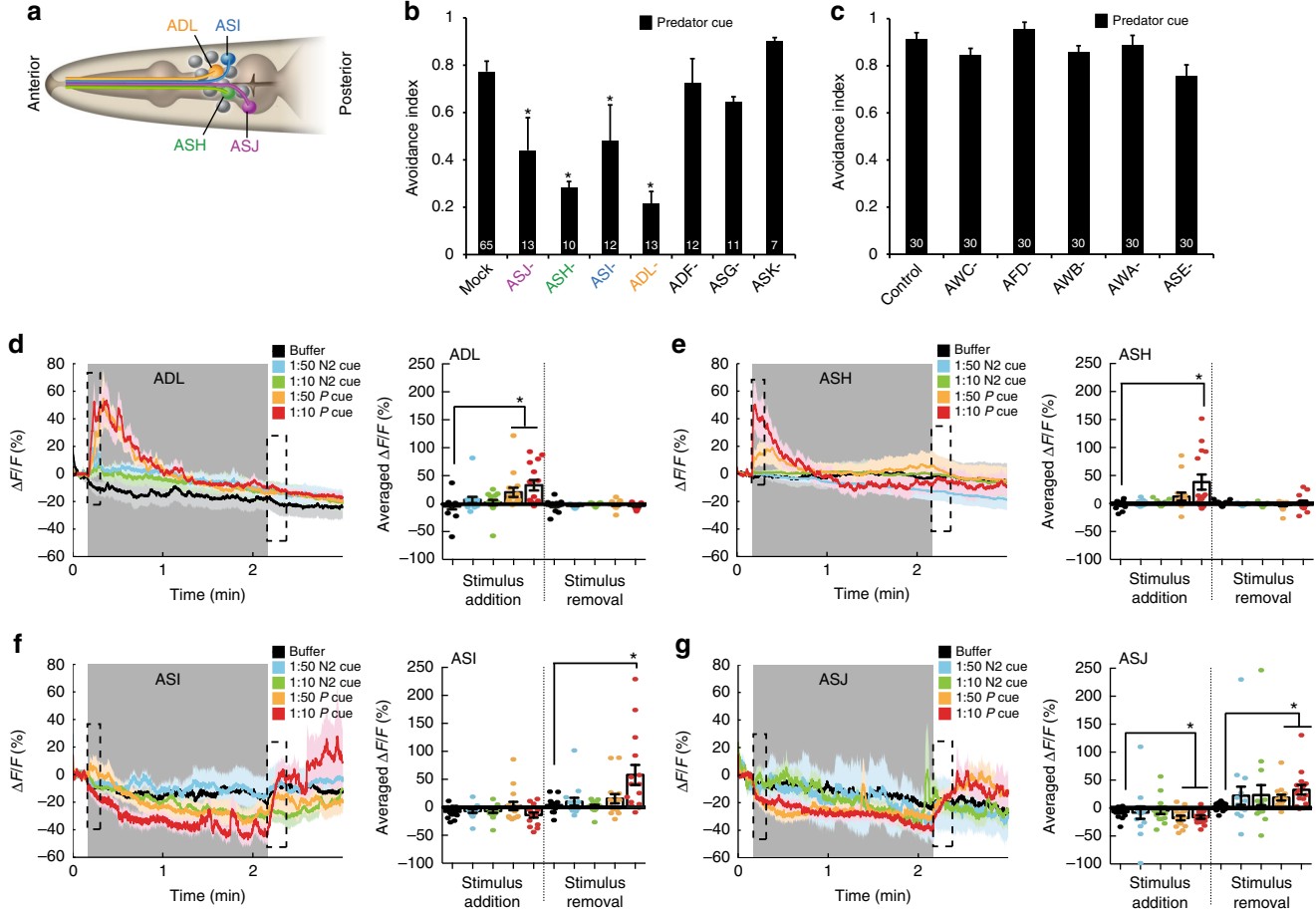

**Fig. 2** Multiple sensory neurons are required for avoidance of predator cue. **a** Schematic showing amphid sensory neurons with key neurons highlighted. **b** Cell and (**c**) genetic ablations showing that ASJ, ASH, ASI, and ADL sensory neurons, but not other amphid neurons, are required for avoidance of predator cue. Averages, s.e.m. and numbers of animals tested are shown on each bar. **d–g** Average calcium responses of transgenic animals ($n > 13$ for each condition) expressing the GCaMP family of indicators in **d** ADL, **e** ASH, **f** ASI, or **g** ASJ sensory neurons to predator cue (*P* cue) or *C. elegans* secretions (N2 cue). Each experiment was a 180 s recording where control (M9 buffer), *C. elegans* secretions (N2 cue), or predator cue (*P* cue) in different dilutions was added at 10 s and removed at 130 s (stimulus is indicated by a shaded gray box). Bar graphs, average percentage change during the 10 s after stimulus addition (dashed box), or 15 s after stimulus removal (dashed box) are shown. Error bars and shaded regions around the curves represent s.e.m. *p < 0.05 obtained by comparison with controls using Fisher's exact t-test with Bonferroni correction

expressing *tax-4* in ASI or ASJ, and *ocr-2* mutants expressing *ocr-2* in ADL or ASH (Fig. 4b). These results indicate that sertraline likely acts downstream of CNG and TRP channels in the ASI, ASJ, ADL and ASH sensory circuits to modulate avoidance responses.

To identify relevant molecular targets of sertraline, we analyzed the behavior of gene mutants in various neurotransmitter-signaling pathways. We found that animals unable to release glutamate (*eat-4*, vesicular glutamate transporter[33]) had reduced responses to predator cue, but showed significant sertraline-induced modulation of avoidance behavior (Fig. 4c), suggesting that glutamate was required for avoidance response, but not for the effect of sertraline on avoidance. In contrast, animals lacking glutamic acid decarboxylase (*unc-25*, encoding an enzyme required for GABA synthesis[34]), but none of the other tested neurotransmitter biosynthetic enzymes were defective in sertraline attenuation (Fig. 4c, Supplementary Fig. S6d). Additionally, adding GABA exogenously to the agar plate was sufficient to restore wild-type-like sertraline-evoked responses in *unc-25* mutants, confirming that GABA signaling is required for sertraline-mediated attenuation of predator (Fig. 4c). These results are consistent with previous studies showing that SSRIs

can modify GABA signaling without affecting serotonin levels in the mammalian brain[35,36], suggesting a broad conservation of the underlying signaling mechanisms.

Next, we tested whether sertraline acts on GABA signaling in specific neurons by restoring UNC-25 function using cell-specific promoters. We found that restoring UNC-25 function to all 26 GABAergic neurons[37] or under a RIS interneuron-selective promoter[38] was sufficient to restore sertraline attenuation of predator avoidance (Fig. 4d). RIS-selective *unc-25* knockdowns showed normal predator avoidance, but reduced sertraline modulation, confirming RIS as the site of sertraline action (Fig. 4d). The RIS interneuron has been previously shown to play a role in inducing a sleep-like state in *C. elegans*[38,39] and our results suggest an additional function for this neuron in modifying predator behavior. Efforts to identify the molecular target of sertraline were unsuccessful as mutants in the vesicular GABA transporter (*unc-47*), an auxiliary transport protein (*unc-46*), the plasma membrane GABA transporter (*snf-11*) or a solute carrier 6 plasma membrane reuptake transporter (*snf-10*) showed normal predator avoidance and sertraline modulation (Fig. 4e), indicating that sertraline acts on an unidentified target in the GABA signaling pathway. Finally, we found that sertraline

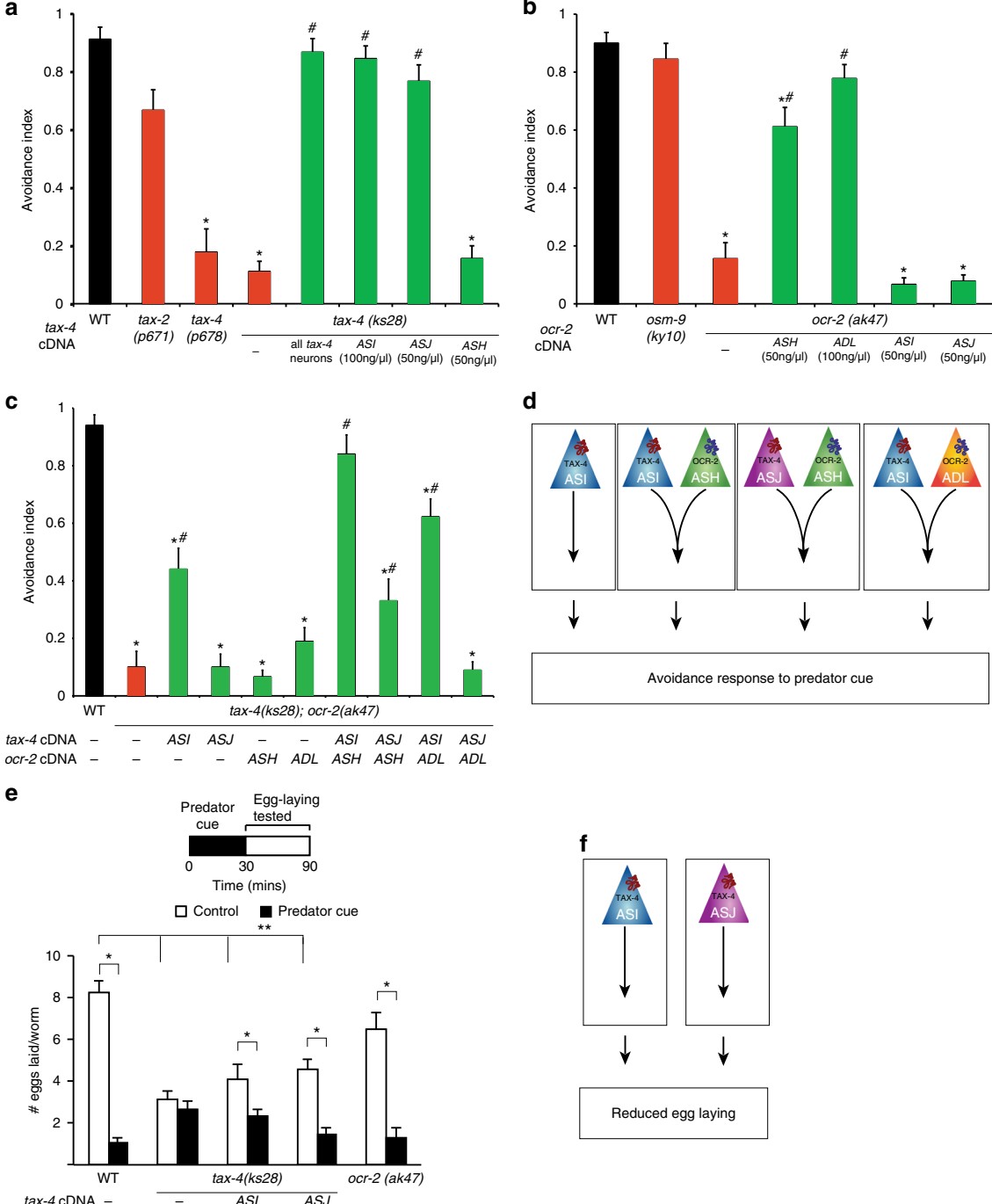

**Fig. 3** A redundant *C. elegans* signaling network enables responses to predator cue. **a** Mutants lacking the alpha subunit of the CNG channel (*tax-4*), but not the beta subunit (*tax-2*), are defective in their response to predator cue. Restoring wild-type *tax-4* cDNA using an ASI- or ASJ-specific promoter is sufficient to restore normal behavior to *tax-4* mutants. **b** Mutants lacking the TRPV channel subunit *ocr-2*, but not *osm-9*, are defective in their response to predator cue. OCR-2 is specifically required in ASH sensory neurons. **c** A *tax-4;ocr-2* double-mutant is also defective in avoiding predator cue and wild-type behavior is restored when TAX-4 is restored to ASI and OCR-2 is restored to ASH neurons. Restoring TAX-4 to ASJ and OCR-2 to ASH or ADL or restoring TAX-4 to ASI and OCR-2 to ADL is able to partially restore wild-type behavior to the double mutants. Moreover, expressing *tax-4* in ASI but not ASJ is sufficient to rescue the double-mutant phenotype. **d** Schematic showing four redundant pathways (ASI acting independently, ASI and ASH, or ASI and ADL, or ASJ and ASH acting together) driving avoidance to *P. pacificus* predator cue. **e** *C. elegans* egg-laying reduction requires functional TAX-4 signaling in ASI and ASJ neurons. OCR-2 is not required for this behavior. **f** Schematic showing that TAX-4 acts in ASI and/or ASJ neurons to reduce egg laying upon longer-term exposure to predator cue. Averages and s.e.m. are shown. **a–c** $n = 30$ animals and in **e,** $n > 35$ tested for each condition, *$p < 0.05$ compared with controls, [#]$p < 0.05$ compared to mutants, **$p < 0.05$ comparing eggs laid by mutants and wild-type animals obtained using Fisher's exact *t*-test with Bonferroni correction

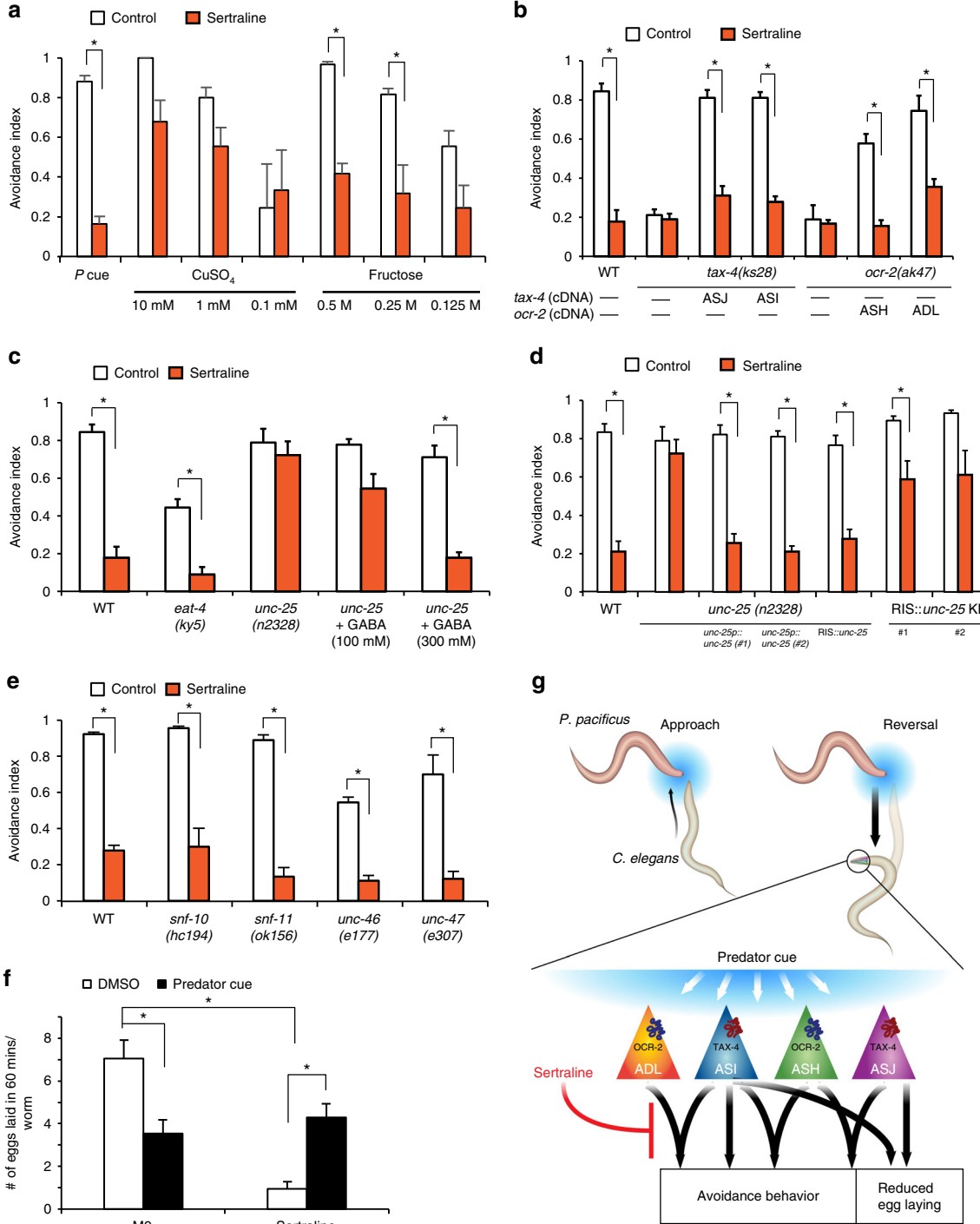

**Fig. 4** Sertraline attenuates *C. elegans* responses to the *P. pacificus* predator. **a** Sertraline attenuates *C. elegans* respnses to predator cue and fructose, whereas copper avoidance is only slightly reduced. **b** The effect of sertraline is lost in *tax-4* and *ocr-2* mutants, and modulation is restored when TAX-4 is restored to either ASI or ASJ, or when OCR-2 is restored to ASH or ADL. **c** Sertraline requires GABA, but not glutamate signaling. Sertraline modulates avoidance responses of *eat-4* (glutamate receptor), but not *unc-25* (glutamic acid decarboxylase, required for GABA synthesis) mutants. Adding GABA exogenously to plates restores sertraline modulation to *unc-25* mutants. **d** Defective *unc-25* response is rescued by expressing wild-type *unc-25* cDNA under the *unc-25* promoter or a RIS-selective promoter. Knocking down *unc-25* specifically in RIS interneuron partially blocks sertraline modulation of predator avoidance. **e** Sertraline attenuates mutants in multiple GABA transporters including *unc-47*, *snf-11*, *snf-10*, and *unc-46* (a transmembrane protein that recruits UNC-47). **f** Animals were treated with predator cue or predator cue and sertraline for 30 min and egg-laying was monitored for 60 min after removal of these compounds. **g** *C. elegans* detects predator cue using sensory circuits consisting of ASI, ASJ, ASH, and ADL neurons that use CNG and TRP channels and act in a redundant manner to generate rapid avoidance. In contrast, CNG channels act in ASI and ASJ neurons to reduce egg laying over many minutes. Sertraline attenuates both predator-induced avoidance behavior and egg-laying behavior downstream of these sensory neurons. Averages of either $n > 30$ (**a–e**) or n > 35 (**f**) and s.e.m. are shown. *$p < 0.05$ compared to controls obtained using Fisher's exact *t*-test with Bonferroni correction (**a–e**) and *$p < 0.05$ compared to controls obtained using a two-way ANOVA (**f**)

treatment also reduced the longer-lasting egg-laying response (Fig. 4f) showing that the drug blocks *C. elegans*' responses on multiple timescales. Taken together, these results indicate that the anti-anxiety drug sertraline abolished predator-induced *C. elegans* responses by acting on GABA signaling in the RIS interneuron.

## Discussion

We show that *P. pacificus* releases a mixture of sulfolipids that *C. elegans* perceives as a predator-specific molecular signature, or kairomone[40]. Perception of these sulfolipids via multiple sensory neurons initiates defensive responses including rapid avoidance and a longer-lasting reduction in egg-laying behavior (Fig. 4g). Among the nematode species whose metabolomes we have analyzed, *P. pacificus* is the only one that excretes copious amounts of sulfolipids. Sulfated fatty acids and related lipids have previously been described primarily from marine sources, including tunicates[41], sponges[42], crustaceans[43], and algae[44]. In addition, a family of sulfated fatty acids, the caeliferins, has been identified from grasshopper oral secretions[45,46]. In a striking parallel to the role of sulfated lipids in the nematode predator-prey system studied here, these herbivore-associated sulfolipids have been shown to elicit specific defense responses in plants[46]. Furthermore, the sulfolipids we identified from *P. pacificus* resemble sodium dodecyl sulfate (SDS), a known nematode repellent[24]. We found that, similar to avoidance triggered by predator cue, ASJ, ASH, and ASI neurons are necessary for avoidance to SDS (Supplementary Fig. S3g). Given the similarity of the neuronal circuitry required for the avoidance responses, it appears that *C. elegans* avoid SDS because of its structural similarity to the *Pristionchus*-released sulfates, which are interpreted as a molecular signature of this predator.

The sulfolipids we identified from *P. pacificus*, sufac#1 and sufal#2, and several related compounds, appear to be derived from the monomethyl branched-chain fatty acid (mmBCFA), C15ISO, which is also produced by *C. elegans* and has been shown to be essential for *C. elegans* growth and development[47]. The biosynthesis of C15ISO in *C. elegans* requires the fatty acid elongase ELO-5, and several homologous elongases in *P. pacificus* exist that may be involved in the biosynthesis of the fatty acid precursors of sufac#1 and sufal#2. Additionally, the biosynthesis of sufac#1 and sufal#2 requires oxygenation at the (ω-5) or (ω-6) position in the fatty acid chain, respectively, followed by sulfation by sulfotransferase(s), a family of genes that has undergone major expansion in *P. pacificus*[48]. Notably, at least one sulfotransferase, EUD-1, functions as a central switch determining whether *P. pacificus* larvae will develop into a primarily bacterivorous, narrow-mouthed adult, or into a predacious, wide-mouthed adult that can feed on other nematodes[49]. It is intriguing that *C. elegans* has evolved the ability to detect a *Pristionchus*-specific trait (the extensive sulfation of small molecules) that is directly connected to the endocrine signaling pathway that controls development of the morphological features required for predation.

Detection of predator cue relies on a sensory neural circuit consisting of at least four different amphid neurons (ASI, ASH, ASJ and ADL, Fig. 4g). These neurons have well-described roles in detecting chemicals from the environment: the ASI and ASJ sensory neurons play a major role in the detection of ascaroside pheromones, whereas the ASH neurons are nociceptive and drive avoidance to glycerol and copper, and ASH and ADL act together to promote social feeding[24,50–52]. Therefore, whereas ASH and ADL have previously been shown to drive avoidance behavior[24,26], our finding that ASI and ASJ are involved in generating avoidance is novel. Participation of these additional neurons facilitates redundancy, such that signaling from ASI, ASI and

ASH, ASI and ADL, ASJ and ASH, and ASJ and ADL is sufficient to drive avoidance to predator cue. Similarly, signaling from either ASI or ASJ neurons alters egg-laying behavior, indicating that these neurons can affect *C. elegans* behavior on multiple timescales. We suggest that ASI and ASJ neurons encode both the removal of predator cue and some aspects of stimulus history. This enables these neurons to drive two behaviors on very different timescales: (1) a rapid avoidance when exposed to predator cue for a second and (2) a 60 min long reduced egg-laying when presented with the same predator cue for 30 min. These results are consistent with previous studies where ASI neurons have been shown to modify long-lasting behaviors[53]. More generally, redundant circuit(s) are likely to decrease the failure rate for signaling, thereby increasing the robustness of the behavioral output. Similar redundant circuits have been described for sensory neurons detecting temperature[54] or odors[55], and in neural circuits driving feeding in the crab[56].

The neuronal signaling machinery in the ASI, ASJ, ASH and ADL sensory neurons relies on CNG and TRP channels to mediate responses to predator cue. CNG ion channels typically consist of alpha and beta subunits and have been shown to play a central role in regulating chemosensory behaviors across multiple species[57,58]. Because the alpha subunit homolog TAX-4 is required for detecting predator cue, whereas the beta subunit TAX-2 is not, we suggest that homomeric TAX-4 channels act in the ASI and ASJ sensory neurons. In vitro experiments have shown that *C. elegans* TAX-4 subunits can form a functional homomeric channel when expressed in HEK293 cells[29]. Similarly, alpha subunits of the CNG channels have also been shown to function as homomeric channels both in vitro and in vivo[59,60]. Our studies also indicate a role for a subunit of the TRP channel OCR-2, but not its heteromeric partner OSM-9[61]. We suggest that OCR-2 can either form a homomeric channel or interact with other non-OSM-9 TRP channel subunits to generate a functional channel and drive avoidance behavior. These results are consistent with previous studies where OCR-2 has been shown to act independently of OSM-9 in regulating *C. elegans* larval starvation[30] and egg-laying behaviors[31]. Our results for the role of TRP channels are reminiscent of rodent studies where TRP channels have been found to play a crucial role in initiating responses to predator odors from cats[3], suggesting broad conservation of the molecular machinery that detects predators. Taken together, we hypothesize that signaling from homomeric CNG and TRP channels acting in distinct, but redundant sensory circuits enable reliable detection of predators by the prey.

We further show that the anti-anxiety drug, sertraline, acts downstream of CNG and TRP channels and requires GABA signaling in RIS interneurons to suppress predator-evoked responses. Sertraline has been shown to be particularly effective in alleviating human anxiety disorders[62] and, classified as an SSRI, is thought to act in part by elevating serotonin levels at synapses[63]. Our studies show that sertraline requires GABA, but not serotonin signaling, to exert its effects on *C. elegans* avoidance behavior. Other SSRIs have also been shown to require GABA signaling in mammalian[35,36] and *C. elegans* nervous systems[64], in addition to effects on other neurotransmitter pathways including dopamine, glutamate, histamine, and acetylcholine[65]. We further show that sertraline action in *C. elegans* requires functional glutamic acid decarboxylase, a GABA biosynthesis enzyme specifically in RIS interneurons, defining the site of action of the drug. RIS interneuron have been implicated in modulating a sleep-like state in *C. elegans*[38]. Previous studies have shown that RIS interneuron released neuropeptide, FLP-11 and not GABA is the major determinant of the sleep-like state[39]. We suggest that GABA signaling in this interneuron might have a role in

modulating *C. elegans* avoidance behaviors, particularly to external threats.

In summary, our results uncover the chemosensory and neuronal basis of a predator-prey relationship between *P. pacificus* and *C. elegans*, in which predator detection is based on a characteristic molecular signature of novel sulfate-containing molecules. The prey uses multiple sensory neurons acting in parallel and conserved CNG and TRP channel signaling to detect these sulfates and drive rapid avoidance and longer-lasting reduced egg-laying responses. Additionally, we show that sertraline acts on GABA signaling in RIS interneurons to attenuate *C. elegans* avoidance behavior. Based on these results, we hypothesize that *C. elegans* evolved mechanisms to detect *Pristionchus*-released sulfolipids as a kairomone, and that the identified neuronal signaling circuitry is representative of conserved or convergent strategies for processing predator threats.

## Methods

**Calcium imaging**. Transgenic animals expressing the genetically encoded calcium indicator, GCaMP[66], under cell-selective promoters were trapped in a custom-designed microfluidic device[27] and their responses to predator cue were recorded. The stimulus (predator cue) was diluted in M9 buffer (3 g $KH_2PO_4$, 6 g $Na_2HPO_4$, 5 g NaCl, 1 ml 1 M $MgSO_4$ per liter of water) as indicated. GCaMP imaging was performed on a Zeiss inverted microscope using a Photometrics EMCCD camera. Images were captured using MetaMorph software at 10 Hz and analyzed offline using MATLAB scripts. Baseline $F_o$ was measured as the average intensity across the first 1–9 s of each recording (or 121–129 s for bar plot quantifications of the responses to stimulus removal). The ratio change in fluorescence to the baseline $F_o$ is plotted in Fig. 2d–g. For bar plots shown in Fig. 2d–g, average responses in a 10 s window after stimulus addition (time 10–20 s) and a 15 s window after removal (time 130–145 s) are shown. Two-tailed unpaired *t*-tests were used to compare neuronal responses, and the Bonferroni correction was used to adjust for multiple comparisons. Individual traces are shown in Figure S4.

**Cell ablations**. For laser ablation experiments, worms were immobilized on agar pads containing 2% agarose with 1 mM sodium azide as anesthetic on glass slides. Individual neurons were identified using DIC optics. Cells were ablated at the L1 larval stage by focusing a laser beam focused through the objective of the microscope. The laser beam is focused in three dimensions on a single spot in the field of view of the identified neuron, specifically the nucleus of the neuron. Pulses of laser were administered and disintegration of the nucleus was monitored under DIC optics. The ablated animals were rescued from the slide and transferred to a standard NGM plate with food. The animals were allowed to recover from the procedure and assayed as young adults 72 h later. Animals with ablated neurons were tested in parallel with controls on the same day.

**Chemotaxis assay**. Chemotaxis assays were performed on 2% agar square (10 cm) plates containing 5 mM potassium phosphate (pH 6), 1 mM $CaCl_2$, and 1 mM $MgSO_4$. Animals were washed three times in M9 and once in chemotaxis buffer (5 mM potassium phosphate (pH 6), 1 mM $CaCl_2$, and 1 mM $MgSO_4$) and then placed in the middle of the plate (Supplementary Fig. 1c). Two spots of the collected nematode secretions and two control M9 spots (all 1 µl) were added to opposite sides of the plate, generating gradients of any volatile components in the nematode secretions. Animals were allowed to explore these gradients for 1 h, after which they were counted and the chemotaxis index calculated (Supplementary Fig. 1d). Benzaldehyde (1:200) and 2-nonanone (undiluted) were used as control attractants and repellents, respectively[25]. Nine or more assays were performed on at least three different days. Two-tailed unpaired *t*-tests were used to compare *C. elegans* responses to different secretions, and the Bonferroni correction was used to adjust for multiple comparisons.

**P. pacificus predator cue**. Nematodes (*C. elegans*, and *P. pacificus* PS312 or RS5275B) were cultured on 10 cm wide NGM plates seeded with lawns of OP50 *E. coli* ($OD_{600} = 0.5$). Animals from 40 such plates were harvested just before the bacteria were completely depleted (6 days), and washed 5 times with M9 buffer. Animals from these plates were pooled into microfuge tubes with approximately 100 µl of worms and the secretions were collected at the starvation times indicated in about 100 µl of M9 buffer. Predator Cue (RS5275B secretions) were tested and diluted in M9 buffer for further experiments. *C. elegans* secretions (N2 cue) were used in Figs 1 and 2, Supplementary Figures S1 and S4. To further concentrate the predator cue for egg-laying assays, we passed the secretions through a Microcon 3 K Centrifugal Filter Column (Millipore), reducing the final volume to 1/10 of the initial volume. This concentrated predator cue was then used in behavioral assays.

**Egg-laying assay**. Synchronized Day 1 adults were treated with M9 control buffer or concentrated predator cue for 30 min. These animals were transferred to a 2-day lawn of 100 µl OP50 bacteria ($OD_{600} = 0.5$) and allowed to lay eggs. Eggs were counted at the time intervals shown. In each condition and genotype, at least 35 animals were analyzed. Average number of eggs and s.e.m. are presented with two-tailed unpaired *t*-tests with Bonferroni correction for statistical analysis. To test the effect of sulfolipids on egg-laying behavior, these compounds were dissolved in DMSO and further diluted in M9 to obtain the concentrations indicated. Sertraline effects were analyzed by exposing animals to predator cue with or without sertraline (1 mM) for 30 min.

**Single animal avoidance assay**. Adult *C. elegans* were exposed to a small volume of a test compound (0.05 µl) near the head of the animal. Upon sensing a repellent, animals initiate a reversal followed by an omega bend. Positive responses to a test compound were scored only if they occurred within 4 s. Results are shown as the avoidance index, which is the ratio of number of positive responses to the total number of trials. Each animal is tested thrice and data is presented as average of the animals tested on at least 3 days (Figs 1–4 and Supplementary Figures S1–S6). For Data in Supplementary Table S5, animals were pre-treated with various drugs at 100 µM for 30 min and then their responses to predator cue were analyzed. Each assay was performed in triplicate (5 animals per condition) and on at least 3 different days ($n < 45$). For Data shown in Fig. 4, animals were pre-treated with 1 mM Sertraline (Sigma) or M9 control for 3 min before being tested for their responses in the avoidance assay. We also performed additional controls to test whether an animal's response to the three repellent trials were independent. We exposed wild-type *C. elegans* to two dilutions of predator cue and SDS and found that in each of these experiments ($n > 24$) the responses were independent (data is presented in Supplementary Table S6), which allowed us to pool our data.

Two-tailed unpaired *t*-tests were used to compare different strains, genotypes, and conditions, and the Bonferroni correction was used to adjust for multiple comparisons. Avoidance indices of all strains tested and controls are shown in Supplementary Table S7.

**Nematode strains and molecular biology**. cDNAs corresponding to the full-length coding sequence for TAX-4 and OCR-2 were obtained from the Bargmann and Liedke labs, respectively. These constructs were sub-cloned under *str-3*, *srh-11*, *sra-6*, and *sre-1* promoters to achieve ASI, ASJ, ASH, and ADL cell-specific expression[67,68]. Transgenic animals were generated by injecting[69] appropriate genotypes with a mixture of the rescuing construct along with co-injection markers as described in Supplementary Table S8.

**Nematode-derived modular metabolite (NDMM) nomenclature**. Nematode metabolites are named using Small Molecule IDentifiers ("SMIDs"), representing searchable, gene-style identifiers that consist of four lower case non-italicized letters followed by a pound sign and a number. The SMID database (www.smid-db.org) is an electronic resource maintained by Profs. Frank C. Schroeder and Lukas Mueller at the Boyce Thompson Institute/Cornell University, in collaboration with Prof. Paul Sternberg at Caltech and WormBase (www.wormbase.org). This database catalogs newly identified nematode small molecules, assigns a unique four-letter SMID (a searchable gene-style identifier), and for each compound includes a list of other names and abbreviations used in the literature.

**Instrumentation for chemical analyses**. NMR spectra were recorded on a Bruker AVANCE III HD (800 MHz) and Varian INOVA-600 (600 MHz) instruments. UHPLC-high-resolution mass spectrometry (HRMS) was performed using a Thermo Scientific Dionex Ultimate 3000 UHPLC system equipped with an Agilent ZORBAX Eclipse XDB C18 column, connected to a Thermo Scientific Q Exactive HF Hybrid Quadrupole-Orbitrap mass spectrometer. HPLC-MS and –MS/MS was performed using an Agilent 1100 Series HPLC system equipped with a diode array detector and an Agilent Eclipse XDB-C18 column (4.6 × 250 mm, 5 µm particle diameter), connected to a Quattro II spectrometer (Micromass/Waters). Flash chromatography was performed using a Teledyne ISCO CombiFlash system. Preparative HPLC separation was performed using the Agilent 1100 Series HPLC system equipped with an Agilent Eclipse XDB-C18 or C8 column (9.4 × 250 mm, 5 µm particle diameter) coupled to a Teledyne ISCO Foxy 200 fraction collector.

**HPLC-MS/MS analyses**. A 0.1% acetic acid water–acetonitrile solvent gradient was used at a flow rate of 1 ml/min, starting with an acetonitrile content of 5% for 5 min and increasing to 100% over a period of 40 min. Exo-metabolome fractions were analyzed by HPLC-ESI-MS in negative and positive ion modes using a capillary voltage of 3.5 kV and a cone voltage of $-35$ V and $+20$ V, respectively. The analytical HPLC protocol mentioned above was translated to a semi-preparative Agilent Eclipse XDB-C18 or C8 column (9.4 × 250 mm, 5 µm particle diameter) with a flow rate of 3.6 ml/min and used for MS-assisted enrichment of desired metabolites, as well as for synthetic sample purification. Data acquisition and processing for the HPLC-MS was controlled by Waters MassLynx software.

**Fig. 5** Overview of synthesis of sufac#1. Reagents and conditions: **a** PCC, DCM, 0 °C; **b** Mg, THF, argon; **c** SO$_3$/pyridine complex, pyridine; **d** NaOH, H$_2$O

**Fig. 6** Overview of synthesis of sufal#1. Reagents and conditions: **a** LiAlH$_4$, THF; **b** *t*-butyldiphenylsilyl chloride (TBDPS-Cl), imidazole, THF; **c** SO$_3$/pyridine complex, pyridine; **d** acetyl chloride, MeOH

**Fig. 7** Overview of synthesis of sufal#2. Reagents and conditions: **a** Pyridinium chlorochromate (PCC), DCM, 0 °C; **b** Mg, THF, argon; **c** LiAlH$_4$, THF; **d** TBDPS-Cl, imidazole, THF; **e** SO$_3$/pyridine complex, pyridine

**UHPLC-HRMS analyses**. A 0.1% formic acid water −0.1% formic acid acetonitrile solvent gradient was used at a flow rate of 0.500 ml/min, starting with an acetonitrile content of 5% for 1.9 min, increasing to 100% over a period of 11 min, and then returning to 5% for 2 min. Data acquisition and processing for the UHPLC-HRMS was controlled by Thermo Scientific Xcalibur software.

***P. pacificus* strains and culture conditions**. RS2333 was used for exo-metabolome preparation. Mixed stage worms from a populated 10 cm NGM agar plate seeded with *E. coli* OP50 were washed into 25 ml of S-complete medium and fed OP50 on days 1, 3 and 5 for a 7-day culture period, while shaking at 22 °C, 220 r.p.m. The cultures were then centrifuged and worm pellets and supernatant frozen separately. For axenic cultures, *P. pacificus* (RS2333) gravid adults from ten 10 cm plates were washed with M9 buffer and treated with alkaline hypochlorite solution to isolate eggs. Isolated eggs were washed thoroughly with M9 buffer and allowed to hatch in fresh sterile M9 for 24 h. The M9 supernatant was prepared as described below.

**Preparation of exo-metabolome extracts and fractionation**
*Protocol A.* *P. pacificus* RS2333 liquid culture supernatant (3 l) was lyophilized to a fine powder and extracted with 750 ml of a 95:5 mixture of ethanol and water for 16 h (2 times). The exo-metabolome extract was then concentrated in vacuo, loaded onto 12 g of ethyl acetate-washed Celite and fractionated using a Teledyne ISCO CombiFlash system and a RediSep GOLD 30 g HP C18 reverse-phase column using a water-methanol solvent gradient, starting with 15 min of 98% water, followed by a linear increase of methanol content up to 100% at 60 min. The eluate was divided into 8 fractions, which were evaporated in vacuo and prepared for HPLC-MS/MS and NMR spectroscopic analyses. Subsequently a subset of fractions was further fractionated by HPLC. A 0.1% acetic acid water–acetonitrile solvent gradient was used at a flow rate of 3.6 ml/min, starting with an acetonitrile content of 5% for 5 min and increasing to 100% over a period of 40 min.

*Protocol B.* *P. pacificus* RS2333 liquid culture supernatant (3 l) was lyophilized to a fine powder and extracted with 3 l of a 95:5 mixture of ethanol and water for 16 h with stirring. The exo-metabolome extract was then concentrated in vacuo, loaded onto 8 g of ethyl acetate-washed Celite and fractionated using a Teledyne ISCO CombiFlash system over a RediSep and GOLD 100 g HP C18 reverse-phase column using a water (0.1% acetic acid)-acetonitrile solvent gradient, starting with 10 min of 100% water, followed by a linear increase of methanol content up to 100% at 92 min, which was maintained up to 100.7 min, thereby producing ~240 fractions, which were prepared for analysis by UHPLC-HRMS.

**Chemical syntheses**
*General methods for chemical synthesis.* Thin-layer chromatography (TLC) was used to monitor progress of reactions unless stated otherwise, using J. T. Baker Silica Gel IB2-F. Unless stated otherwise, reagents were purchased from Sigma-Aldrich and used without further purification. *N,N*-dimethylformamide (DMF)

and dichloromethane (DCM) were dried over 4 Å molecular sieves prior to use. Tetrahydrofuran (THF) was distilled over lithium aluminum hydride prior to use. Optical rotations were measured on a Perkin Elmer 341 polarimeter. Synthetic schemes are shown in Figs 5, 6, 7.

**Preparation of 2 (methyl 10-oxo-decanoate)**. **1** (2.5 mL, 13.5 mmol) was added to a solution of pyridinium chlorochromate (2.91 g, 13.51 mmol) and DCM (50 ml) at 0 °C. After stirring for 20 min, the reaction was quenched with ether (30 ml), filtered over Celite, and washed further with ether (20 ml). The filtrate was concentrated in vacuo. Flash column chromatography on silica using a gradient of 0–100% ethyl acetate in hexanes afforded **2** at 52% purity (48% dimer) (226.12 mg, 1.13 mmol, 8%). $^1$H NMR (400 MHz, chloroform-*d*): δ (p.p.m.) 9.76 (t, *J* = 2.0 Hz, 1 H), 3.56 (s, 3 H), 2.42 (td, *J* = 7.4, 1.6 Hz, 2 H), 2.30 (t, 7.6 Hz, 2 H), 1.55–1.48 (m, 4 H), 1.25–1.19 (m, 10 H).

**Preparation of 3 (methyl 10-hydroxy-13-methyltetradecanoate)**. Ethylene dibromide (2 drops, 0.58 mmol) was added to Mg turnings (729 mg, 30 mmol) in THF (5 ml) under argon and the mixture was heated briefly until reflux. After cooling to room temperature, a solution of 3-methylbromobutane (1.2 ml, 10 mmol) in THF (10 ml) was added dropwise to the reaction over 10 min with stirring. The reaction was stirred at reflux for 1 h, affording isopentylmagnesium bromide. The isopentylmagnesium bromide solution (100 µl, 0.9 mmol) was then added to a solution of **2** (175 mg, 874 µmol) in THF (1 ml) at −25 °C. The reaction was allowed to return to room temperature while stirring, at which point the reaction was quenched with saturated NH$_4$Cl (10 ml) and extracted with hexanes (10 ml). Flash column chromatography on silica using a gradient of 0–100% ethyl acetate in hexanes afforded **3** at 70% purity (90.6 mg, 333 µmol, 38%). $^1$H NMR (600 MHz, chloroform-*d*): δ (p.p.m.) 3.66 (s, 3 H), 3.55 (m, 1 H), 2.30 (t, 7.7 Hz, 2 H), 1.64–1.32 (m, 9 H), 1.20 (m, 1 H), 1.30 (m, 10 H), 0.90 (d, 6.6 Hz, 3 H), 0.89 (d, 6.6 Hz, 3 H).

**Preparation of 4 (13-methyl-10-(sulfooxy)tetradecanoic acid) (sufac#1)**. Sulfur trioxide-pyridine complex (120 mg, 0.75 mmol) was added to a solution of **3** (70 mg, 257 µmol) and pyridine (1 ml). After stirring for 5 min, the solution was concentrated in vacuo and dissolved in NaOH (aq) (4 ml, 2.75 M). The reaction was then neutralized with acetic acid/TFA to a pH of 4 and concentrated in vacuo over Celite. Flash column chromatography on silica using a gradient of 0–100% methanol in DCM afforded **4** (9.6 mg, 30 µmol, 12%). Spectroscopic data were in agreement with those for the natural product (see Supplementary Tables S2 and S3).

**Preparation of 6 (14-((*t*-butyldiphenylsilyl)oxy)-2-methyltetradecan-5-ol)**. TBDPSCl (30 µL, 0.11 mmol, ~3 eq) was added incrementally over the course of 24 h to a solution of **5** (10.2 mg, 40 µmol) and imidazole (3.51 mg, 0.13 mmol, ~3 eq) in THF (2 ml). The reaction was concentrated in vacuo. Flash column chromatography on silica using a gradient of 0–100% ethyl acetate in hexanes

afforded **6** at 54% purity (6.04 mg, 12.96 μmol, 32.4%). ¹H NMR (600 MHz, methanol-$d_4$): δ (p.p.m.) 7.72 (m, 4 H), 7.36 (m, 6 H), 3.66 (t, 6.4, 2 H), 3.48 (m, 1 H), 1.57–1.25 (m, 20 H), 1.2 (m, 1 H), 1.03 (s, 9 H), 0.90 (d, 6.6 Hz, 3 H), 0.89 (d, 6.6 Hz, 3 H).

**Preparation of 7 (14-((t-butyldiphenylsilyl)oxy)-2-methyltetradecan-5-yl hydrogen sulfate).** Sulfur trioxide-pyridine complex (excess) was added to a solution of **6** (6.04 mg, 12.96 μmol) in pyridine (1 ml) and stirred for 5 min at room temperature. The reaction was concentrated in vacuo. Flash column chromatography on silica using a gradient of 0–100% methanol in DCM afforded **7** at 20% purity (3.3 mg, 6 μmol, 46% yield). ¹H NMR (600 MHz, methanol-$d_4$): δ (p.p.m.) 7.72 (m, 4 H), 7.36 (m, 6 H), 4.31 (quin, 6.08 Hz, 1 H), 3.66 (t, 6.4 Hz, 2 H), 1.70–1.60 (m, 4 H), 1.6–1.5 (m, 3 H), 1.47–1.22 (m, 14 H), 1.03 (s, 9 H), 0.90 (d, 6.6 Hz, 3 H), 0.89 (d, 6.6 Hz, 3 H).

**Preparation of 8 (14-hydroxy-2-methyltetradecan-5-yl hydrogen sulfate) (sufal#1).** Methanol (500 μl) with a catalytic amount of acetyl chloride (0.5 μl) was added to a solution of **7** (1.3 mg, 2.3 μmol). The solution was concentrated in vacuo. Flash column chromatography on silica using a gradient of 0–100% methanol in DCM afforded **8** (648 μg, 2.0 μmol, 87% yield). Spectroscopic data were in agreement with those for the natural product: ¹H NMR (600 MHz, methanol-$d_4$): δ (p.p.m.) 4.31 (quin, 6.1 Hz, 1 H), 3.54 (t, 6.8 Hz, 2 H), 1.7–1.58 (m, 5 H), 1.57–1.49 (m, 3 H), 1.47–1.23 (m, 13 H), 0.91 (d, 6.6 Hz, 3 H), 0.90 (d, 6.6 Hz, 3 H); ¹³C NMR (800 MHz, methanol-$d_4$): δ (p.p.m.) 62.76, 33.40, 26.64, 30.28, 30.xx, 30.xx, 30.xx, 25.75, 35.00, 80.85, 32.91, 34.87, 28.98, 22.68, 22.68

**Preparation of 10 (methyl-9-oxo-nonanoate).** **9** (600 μl, 2.8 mmol) was added to a solution of PCC (900 mg, 4.2 mmol) in dry DCM (50 ml) under argon at −15 °C, and the mixture was stirred for 45 min. Reaction progress was monitored by TLC. The reaction mixture was allowed to return to room temperature and was stirred for an additional 40 min. The mixture was then filtered over Celite, and the residue was washed with ether. The filtrate was concentrated in vacuo. Flash column chromatography on silica using a gradient of 0–100% ethyl acetate in hexanes afforded **10** at 70% purity (containing about 30% starting material) (305 mg, 1.64 mmol, 58% yield). ¹H NMR (600 MHz, chloroform-$d$): δ (p.p.m.) 9.76 (t, 1.79 Hz, 1 H), 3.66 (s, 3 H), 2.42 (td, 7.30, 1.77 Hz, 2 H), 2.30 (t, 7.60 Hz, 2 H), 1.62 (m, 4 H), 1.32 (m, 6 H).

**Preparation of 11 (methyl 9-hydroxy-13-methyltetradecanoate).** Ethylene dibromide (6 drops, 1.74 mmol) were added to Mg turnings (500 mg, 20.6 mmol, ~2 eq) in THF (5 ml) under argon. And the mixture was heated to reflux for 20 min. After cooling to room temperature, 4-methylbromopentane (1.38 ml, 10 mmol) in THF (10 ml) was added quickly to the solution of activated Mg turnings. The mixture was then refluxed for 1 h. After cooling to room temperature, the Grignard reagent (1.36 ml, 0.9 mmol, 1 eq) was added to a solution of **10** (170 mg, 0.9 mmol) in THF (1 ml). The reaction was monitored by TLC using 1:4 acetone:hexanes. The reaction was quenched with saturated NH₄Cl (10 ml) and extracted with hexanes. Flash column chromatography on silica using a gradient of 0–100% ethyl acetate in hexanes afforded **11** at 60% purity (81.7 mg, 0.3 mmol, 32%). ¹H NMR (400 MHz, chloroform-$d$): δ (p.p.m.) 3.66 (s, 3 H), 3.58 (m, 1 H), 2.30 (t, 7.58 Hz, 2 H), 1.65–1.49 (m, 4 H), 1.47–1.36 (m, 4 H), 1.31 (m, 9 H), 1.16 (m, 1 H), 0.88 (d, 6.64 Hz, 3 H), 0.88 (d, 6.64 Hz, 3 H).

**Preparation of 12 (13-methyltetradecane-1,9-diol).** **11** (81.7 mg, 300 μmol) was added dropwise to a suspension of lithium aluminum hydride (102.3 mg, 2.69 mmol) in THF (20 ml) at 0 °C. The solution was warmed slowly to room temperature while stirring for 20 min. The reaction was quenched by the Fieser method as described above, filtered over Celite, and concentrated in vacuo. Flash column chromatography on silica using a gradient of 0–100% ethyl acetate in hexanes afforded **12** at 90% purity (57.0 mg, 234 μmol, 77.4%). ¹H NMR (400 MHz, methanol-$d_4$): δ (p.p.m.) 3.54 (t, 6.53 Hz, 2 H), 3.50 (m, 1 H), 1.61–1.13 (m, 21 H), 0.89 (d, 6.6 Hz, 3 H), 0.89 (d, 6.6 Hz, 3 H).

**Preparation of 13 (14-((tert-butyldiphenylsilyl)oxy)-2-methyltetradecan-6-ol).** TBDPS-Cl (97.6 μl, 370 μmol) was added incrementally over the course of 3 h to a solution of **12** (57.0 mg, 234 μmol) in THF (1 ml) and DMF (0.5 ml). The reaction was concentrated in vacuo. Flash column chromatography on silica using a gradient of 0–100% ethyl acetate in hexanes afforded **13** at 30% purity (13.1 mg, 23.4 μmol, 10%). ¹H NMR (400 MHz, methanol-$d_4$): δ (p.p.m.) 7.67–7.64 (m, 4 H), 7.41–7.35 (m, 6 H), 3.66 (t, 6.1 Hz, 2 H), 3.50 (m, 1 H), 1.61–1.13 (m, 21 H), 1.03 (s, 9 H), 0.89 (d, 6.6 Hz, 3 H), 0.89 (d, 6.6 Hz, 3 H).

**Preparation of 14 (14-hydroxy-2-methyltetradecan-6-yl hydrogen sulfate) (sufal#2).** Sulfur trioxide-pyridine complex (50 mg, excess) was added to a solution of **13** (20 μmol) in acetonitrile (2 ml). The solution was concentrated in vacuo. Flash column chromatography on silica using a gradient of 0–100%

methanol in DCM afforded **14** (0.65 mg, 2 μmol, 10%). Spectroscopic data were in agreement with those for the natural product (see Supplementary Tables S1 and S3).

**Data availability.** Data presented in this manuscript is either shown in Supplementary Figures or Supplementary Tables or maintained with the authors. Sulfo-lipid structures have been deposited to the Metabolights database, Study Identifier MTBLS611 (https://www.ebi.ac.uk/metabolights/MTBLS611).

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

## Acknowledgements

We thank R. Hong, R. Sommer, C. Bargmann, S. Mitani, the National BioResource Project and Caenorhabditis Genetics Center (CGC) for strains; C. Bargmann and W. Liedke for constructs; and P. Sengupta, A. Chisholm, R. Hong, C. Bargmann, Y. Jin, D. O'Keefe, K. Quach, U. Magaram, H. Lau, L. Hale, and other members of the Chalasani Laboratory for helpful discussions and comments on the manuscript. This work was supported by grants from the W. M. Keck Foundation, NIH R01MH098001, R01MH113905 (S.H.C.), NIH R01AT008764 (F.C.S.), a Salk Alumni Fellowship (K.P.C) and NIH R01DC016058 (J.S.). A.K.P. holds a graduate research fellowship from the National Science Foundation.

## Author contributions

Z.L., M.J.K., C.D.C., A.K.P., S.G.L., A.T., K.P.C., and N.B. performed experiments, developed experimental methods and reagents. F.C.S., J.S., and S.H.C. designed and interpreted the experiments and wrote the paper.
