## [Peer Review File(PDF 341 kb) · Nature Communications]

Reviewers' comments:

Reviewer #1 (Remarks to the Author):

The overall premise of this paper, and its primary finding, are quite interesting, novel, and convincing. Liu and colleagues discover that the *c. elegans* predator *Pristionchus pacificus* releases a suite of sulfolipids upon starvation, and that these molecules serve as a potent avoidance cue for *c. elegans*. Cell ablations show that there are four sensory neurons required for normal avoidance of these cues, and that two of them (ADL, ASH) show responses to sulfolipids, while the other two (ASH and ASJ) show responses to the removal of the cue. Moreover, a series of genetic experiments demonstrates that the CNG channel TAX-4 is required in the "off" neurons while the TRP channel *ocr-2* is required in the "on" neurons.

I was less compelled by the last element of the paper, which was the demonstration that the anti-anxiety drug Sertraline, an SSRI, blocks the avoidance response by acting through GABAergic neurons. I have a couple reservations about this conclusion:

1) Figure 1a is intended to demonstrate the reduction in predator avoidance in response to Sertraline, as well as the specificity of this effect. However, I'm left unconvinced that this is a specific effect on predator avoidance, and not a generalized reduction in locomotion or avoidance behavior (sluggish worms). All three controls (0.5 M Fructose, 5M NaCl, 10 mM CuSO₄) induce incredibly strong avoidance, to a level that could be considered saturated. Thus, the more modest reduction in avoidance of these stimuli following Sertraline treatment may be an intensity effect, where even "sluggish" worms avoid these stimuli because they are so intensely aversive. To convincingly demonstrate specificity to predator cue avoidance, a control cue inducing a similar level of avoidance should be tested for the effect of Sertraline.

2) The authors state that Sertraline requires GABA signaling in RIS neurons for its effect, but this is not formally demonstrated. The RIS-specific *unc-25* rescue certainly demonstrates that GABA signaling restricted to RIS is sufficient to mediate the effects of Sertraline on reducing avoidance. However, it is possible that selective restoration in other GABAergic neurons would have the same effect. Cell-specific knockdown of GABA signaling in RIS is required for the stated conclusion. Otherwise, the conclusion should be modified to better reflect the data.

3) The *snf-10* result should be discussed in more detail from a molecular perspective. For example, if Sertraline is thought to potentially act through *snf-10* by inhibiting it, one would have expected the *snf-10* mutants themselves to show reduced avoidance. Since they do not, the alternative is that Sertraline increases the function of *snf-10*, but would this be consistent with the cited interactions seen with mammalian homologs?

Additionally, I have some more general comments/concerns.

4) The statistical tests used need to be better described and justified. In particular, the

Fisher's test is used throughout, but some details of its use are missing. For example, each animal is tested three times for avoidance. Is each trial counted as an independent replicate? The use of the Fisher's test suggests this is the case; however, there is obviously pseudoreplication present if each trial (rather than each animal) is used, which arguably biases the Fisher's test. And does the reported "n" reflect the number of trials or the number of animals? (note: n is not reported for some figures, so please add this information). Also, if the data is binary, the use of s.e.m. is strange to me, and should be justified. Reporting the standard error on 1s and 0s doesn't seem particularly informative, and will largely simply reflect the sample size.

5) Another comment on the statistics: In figure 4e, the # sign is not explained in the legend. I assume this means that the Sertraline treated value is significantly different from the control Sertraline response, but this should be clarified. Also, in a two-factor analysis (genotype and Sertraline treatment), arguing that the Sertraline treatment affects two genotypes differently requires an analysis of interaction between those two factors. It is not clear whether this is done. Similarly, in figure 4f, it is argued that Sertraline treatment reduces the effect of predator cue on egg laying reduction. However, as far as I can tell, each value is only compared to controls. Finding that Sert+cue is not significantly different from the controls, whereas cue alone is significantly different from controls, is not a logically valid argument for Sert+cue being different from cue alone. This difference must be tested directly.

6) minor:

- on line 141, should "ASJ" be "ADL"?
- line 177: Fig. S6c should be S6d.

Reviewer #2 (Remarks to the Author):

Predator-secreted sulfolipids induce fear-like defensive responses in *C. elegans*. The authors show that *C. elegans* avoids the excretion of starved *P. pacificus* nematodes upon contact and that exposure to the predator cue delays egg-laying. These responses require sensory neurons ASJ, ASH, ASI, or ADL, a CNG channel and a TRP channel in subsets of these neurons direct the behavioral responses. The authors also identify that several (ω -1)-branched-chain sulfolipids are produced by starved *P. pacificus* nematodes and synthesized sulfates generate similar responses in *elegans* worms. In addition, sertraline attenuates a worm's response to the predator cue by regulating GABA signaling in the neuron RIS. This study characterizes the cue, the sensory neurons, the cellular events and the potential downstream signals that regulate the response of *C. elegans* to starved *P. pacificus*. It provides new understanding of predator-prey interaction. However, several interesting results need to be clarified.

1. They show that combining the function of a CNG and a TRP in subsets of the 4 sensory neurons directs avoidance. Meanwhile, they also show that the function of the CNG and the TRP depends on the concentration of the expressing plasmids. One possibility is that the specific combinations that they found is a result of different dosages.

2. The predator cue generates both immediate avoidance and a long-term effect on egg laying. However, the authors did not explore at all how these two responses of different time scales are regulated by similar neurons. The potential mechanisms should be at least well discussed.
3. RIS regulates sleep. They did not provide any insight into how sertraline acts on RIS to attenuate the response to the predator cue.

Reviewer #3 (Remarks to the Author):

This paper describes a newly-identified predator-avoidance behavior in *C. elegans*. Specifically, when *C. elegans* encounter chemical traces of the predatory nematode *Pristionchus pacificus*, they exhibit avoidance behavior similar to that previously observed for chemical repellents. The authors identify sulfolipid released by starved *Pristionchus* that are responsible for predator avoidance. They also identify neurons, and sensory ion channels required in these neurons, for detection of predator-indicating sulfolipids. Finally, they show that sertraline, an anti-anxiety drug, suppresses predator avoidance behavior, a process requiring GABA release from the RIS interneurons.

This is an interesting study that defines a novel behavior of ecological and neuropharmacological interest. The authors make a good start to discover the underlying neural and molecular mechanisms of this process. In principle I think the paper is appropriate for *Nature Communications*.

One aspect of the paper that seemed slightly incomplete was the connection between the response to predator cues and their suppression by sertraline. The RIS neuron, which the authors identify as the focus of this suppression, is at least two synapses downstream of the neurons that sense the cues. So while I can understand the authors' implicit suggestion that sertraline is suppressing the worms' anxiety about predators, it seems also plausible that potentiation of GABA signaling from RIS is generally suppressing avoidance responses. It would be straightforward for the authors to address this by testing whether sertraline/RIS GABA signaling affects responses to other repellents sensed by the ASH, ADL, etc neurons.

Regarding the phenotypes of *tax-4* and *ocr-2* mutants: the authors propose that TAX-4 functions as a homomeric channel, but there are other cng genes (*cng-1* and *cng-2*) which could heteromerize with *tax-4* in ASI and ASJ. It might be worthwhile to test these in their assays.

Also, I think this sentence "Also, since TAX-4 but not TAX-2 can form a homomeric CNG channel²⁸, we hypothesize that the OCR-2 TRP channel..." (on p 5) is a mistake.

Reviewer #1 (Remarks to the Author):

The overall premise of this paper, and its primary finding, are quite interesting, novel, and convincing. Liu and colleagues discover that the *c. elegans* predator *Pristionchus pacificus* releases a suite of sulfolipids upon starvation, and that these molecules serve as a potent avoidance cue for *c. elegans*. Cell ablations show that there are four sensory neurons required for normal avoidance of these cues, and that two of them (ADL, ASH) show responses to sulfolipids, while the other two (ASH and ASJ) show responses to the removal of the cue. Moreover, a series of genetic experiments demonstrates that the CNG channel TAX-4 is required in the “off” neurons while the TRP channel ocr-2 is required in the “on” neurons.

I was less compelled by the last element of the paper, which was the demonstration that the anti-anxiety drug Sertraline, an SSRI, blocks the avoidance response by acting through GABAergic neurons. I have a couple reservations about this conclusion:

1) Figure 1a is intended to demonstrate the reduction in predator avoidance in response to Sertraline, as well as the specificity of this effect. However, I’m left unconvinced that this is a specific effect on predator avoidance, and not a generalized reduction in locomotion or avoidance behavior (sluggish worms). All three controls (0.5 M Fructose, 5M NaCl, 10 mM CuSO₄) induce incredibly strong avoidance, to a level that could be considered saturated. Thus, the more modest reduction in avoidance of these stimuli following Sertraline treatment may be an intensity effect, where even “sluggish” worms avoid these stimuli because they are so intensely aversive. To convincingly demonstrate specificity to predator cue avoidance, a control cue inducing a similar level of avoidance should be tested for the effect of Sertraline.

Response: We thank the reviewer for their positive comments. We agree that the initially tested concentrations of the controls (0.5 M Fructose, 5 M NaCl and 10 mM CuSO₄) induced very strong avoidance. We have now tested multiple dilutions of these repellents and find that sertraline attenuates avoidance of different agents to different extents. These data are presented in Supplementary Figure S6. We have modified the manuscript text accordingly.

2) The authors state that Sertraline requires GABA signaling in RIS neurons for its effect, but this is not formally demonstrated. The RIS-specific *unc-25* rescue certainly demonstrates that GABA signaling restricted to RIS is sufficient to mediate the effects of Sertraline on reducing avoidance. However, it is possible that selective restoration in other GABAergic neurons would have the same effect. Cell-specific knockdown of GABA signaling in RIS is required for the stated conclusion. Otherwise, the conclusion should be modified to better reflect the data.

Response: We have knocked down *unc-25* specifically in RIS neurons and found that these transgenic animals showed reduced sertraline modulation (consistent with the reviewer’s prediction). This data is presented in Figure 4.

3) The *snf-10* result should be discussed in more detail from a molecular perspective. For example, if Sertraline is thought to potentially act through *snf-10* by inhibiting it, one would have expected the *snf-10* mutants themselves to show reduced avoidance. Since they do not, the alternative is that Sertraline increases the function of *snf-10*, but would this be consistent with the cited interactions seen with mammalian homologs?

Response: We re-tested *snf-10* mutants and found that attenuation by sertraline in this mutant is only slightly weaker than in control and the other tested mutants. The combined data don't show significance, and we have modified the discussion accordingly.

Additionally, I have some more general comments/concerns.

4) The statistical tests used need to be better described and justified. In particular, the Fisher's test is used throughout, but some details of its use are missing. For example, each animal is tested three times for avoidance. Is each trial counted as an independent replicate? The use of the Fisher's test suggests this is the case; however, there is obviously pseudoreplication present if each trial (rather than each animal) is used, which arguably biases the Fisher's test. And does the reported "n" reflect the number of trials or the number of animals? (note: n is not reported for some figures, so please add this information). Also, if the data is binary, the use of s.e.m. is strange to me, and should be justified. Reporting the standard error on 1s and 0s doesn't seem particularly informative, and will largely simply reflect the sample size.

Response: We agree with the reviewer that the description of the statistical tests was incomplete. We have now included additional information and re-did all of our statistical tests. For each experiment, animals were tested thrice and the number of animals (and not number of trials) were used for statistics. Further, we found that animals showed similar responses to both predator and SDS for all three trials, suggesting that these responses are independent. Data presented in the figures was obtained over multiple days (with at least 3 animals tested on each day) and the s.e.m. is a measure of variance observed. Detailed descriptions and results can be found in Methods section and the Supplementary Materials.

5) Another comment on the statistics: In figure 4e, the # sign is not explained in the legend. I assume this means that the Sertraline treated value is significantly different from the control Sertraline response, but this should be clarified. Also, in a two-factor analysis (genotype and Sertraline treatment), arguing that the Sertraline treatment affects two genotypes differently requires an analysis of interaction between those two factors. It is not clear whether this is done. Similarly, in figure 4f, it is argued that Sertraline treatment reduces the effect of predator cue on egg laying reduction. However, as far as I can tell, each value is only compared to controls. Finding that Sert+cue is not significantly different from the controls, whereas cue alone is significantly different from controls, is not a logically valid argument for Sert+cue being different from cue alone. This difference must be tested directly.

Response: We have re-tested *snf-10* mutants and re-analyzed this data using a one-way ANOVA, and the combined data no longer show a significant effect of *snf-10* on avoidance attenuation by sertraline.

6) minor:

- on line 141, should "ASJ" be "ADL"?

- line 177: Fig. S6c should be S6d.

Response: This has been corrected. Thanks!

Reviewer #2 (Remarks to the Author):

Predator-secreted sulfolipids induce fear-like defensive responses in *C. elegans*

The authors show that *C. elegans* avoids the excretion of starved *P. pacificus* nematodes upon contact and that exposure to the predator cue delays egg-laying. These responses require sensory neurons ASJ, ASH, ASI, or ADL, a CNG channel and a TRP channel in subsets of these neurons direct the behavioral responses. The authors also identify that several (ω -1)-branched-chain sulfolipids are produced by starved *P. pacificus* nematodes and synthesized sulfates generate similar responses in *C. elegans* worms. In addition, sertraline attenuates a worm's response to the predator cue by regulating GABA signaling in the neuron RIS. This study characterizes the cue, the sensory neurons, the cellular events and the potential downstream signals that regulate the response of *C. elegans* to starved *P. pacificus*. It provides new understanding of predator-prey interaction. However, several interesting results need to be clarified.

1. They show that combining the function of a CNG and a TRP in subsets of the 4 sensory neurons directs avoidance. Meanwhile, they also show that the function of the CNG and the TRP depends on the concentration of the expressing plasmids. One possibility is that the specific combinations that they found is a result of different dosages.

Response: We find that CNG signaling from either ASI or ASJ is sufficient to drive predator avoidance behavior. Similarly, we observe that TRP signaling from ASH or ADL can drive predator avoidance. We agree that it is possible that we observe these combinations to be sufficient as a result of the different dosages. We have included this possibility in our text.

2. The predator cue generates both immediate avoidance and a long-term effect on egg laying. However, the authors did not explore at all how these two responses of different time scales are regulated by similar neurons. The potential mechanisms should be at least well discussed.

Response: We have added further discussion about the potential mechanisms that drive these behaviors that occur on multiple timescales.

3. RIS regulates sleep. They did not provide any insight into how sertraline acts on RIS to attenuate the response to the predator cue.

Response: We have included further discussion into the role of RIS, sleep and Sertraline.

Reviewer #3 (Remarks to the Author):

This paper describes a newly-identified predator-avoidance behavior in *C. elegans*. Specifically, when *C. elegans* encounter chemical traces of the predatory nematode *Pristionchus pacificus*, they exhibit avoidance behavior similar to that previously observed for chemical repellents. The authors identify sulfolipid released by starved *Pristionchus* that are responsible for predator avoidance. They also identify neurons, and sensory ion channels required in these neurons, for detection of predator-indicating sulfolipids. Finally, they show that sertraline, an anti-anxiety drug, suppresses predator avoidance behavior, a process requiring GABA release from the RIS interneurons.

This is an interesting study that defines a novel behavior of ecological and neuropharmacological interest. The authors make a good start to discover the underlying neural and molecular mechanisms of this process. In principle I think the paper is appropriate for Nature

Communications.

One aspect of the paper that seemed slightly incomplete was the connection between the response to predator cues and their suppression by sertraline. The RIS neuron, which the authors identify as the focus of this suppression, is at least two synapses downstream of the neurons that sense the cues. So while I can understand the authors' implicit suggestion that sertraline is suppressing the worms' anxiety about predators, it seems also plausible that potentiation of GABA signaling from RIS is generally suppressing avoidance responses. It would be straightforward for the authors to address this by testing whether sertraline/RIS GABA signaling affects responses to other repellents sensed by the ASH, ADL, etc neurons.

Response: We have included additional evidence to show that sertraline does attenuate responses to lower dilutions of Fructose and to a lesser extent, CuSO₄. These results indicate that sertraline might have a broader effect on avoidance behavior. Also, we find that RIS GABA signaling is only required for sertraline modulation and not predator avoidance alone, indicating these two pathways can be separated. We have modified our text to discuss these results.

Regarding the phenotypes of *tax-4* and *ocr-2* mutants: the authors propose that TAX-4 functions as a homomeric channel, but there are other *cng* genes (*cng-1* and *cng-2*) which could heteromerize with *tax-4* in ASI and ASJ. It might be worthwhile to test these in their assays.

Response: We agree that additional *cng* genes might heteromerize with *tax-4* in ASI and ASJ neurons and we now have included this possibility in our discussion. However, *cng-1* and *cng-2* are both alpha subunits, similar to TAX-4 and might not heteromerize with TAX-4. Typically, alpha subunits interact with beta subunits to form functional channels. However, *tax-4* has been found to form homomeric channels *in vitro*. Collectively, we speculate that *tax-4* can function by itself in ASI and ASJ neurons.

Also, I think this sentence "Also, since TAX-4 but not TAX-2 can form a homomeric CNG channel²⁸, we hypothesize that the OCR-2 TRP channel..." (on p 5) is a mistake.

Response: We have revised this sentence, thanks!

REVIEWERS' COMMENTS:

Reviewer #1 (Remarks to the Author):

After reading the revised manuscript and the responses to all the reviewer comments, my opinion of the paper remains largely unchanged. The identification of the predator cue and the sensory neuron/genetic mechanisms of sensing and reacting to this cue are all very nice and appear robust. The impact of sertraline on avoidance, however, remains less satisfying. In fact, I think it's fair to say that the evidence for sertraline acting specifically on predator avoidance is actually weaker now that two previous conclusions have been reversed (that sertraline does not affect avoidance of other cues, and that *snf-10* mutants retain sertraline modulation.) As an aside, the fact that two repeated experiments failed to replicate raises some concerns about the experiments that were not repeated, but I'm willing to accept that this may be a low probability event and the other conclusions are sound.

I would be more inclined to support publication based on the identification of the predator derived cue and the sensory detection of that cue, if it were not for the fact that the sertraline result, and the implication that this suggests some conserved (or convergent) mechanism for stress/anxiety-driven predator responses appears to remain a cornerstone of the paper. I fear that the publicised narrative will be that worms get stressed by predators and their anxiety can be relieved with antidepressants. While this may be true, the evidence that there is a "conserved or convergent strategy for managing predator threats" is weak. This is a drug that may have all kinds of effects on different neurotransmitter systems. The fact that it (nonspecifically) reduces avoidance to a cue that happens to be secreted by a predator is not great evidence for functional conservation.

If the effects of sertraline and the idea of functional conservation were presented as more preliminary and tentative, I would be more supportive of the paper. Whether this weakens the impact is a question that is probably best dealt with editorially.

Two more minor comments relating to the revision:

1) On line 195 it is stated that sertraline "affects some, but not all repellent circuits". Given that it affects all three cues tested, it's not clear to me how the "some, but not all" part of this statement is justified.

2) The figure legend for Figure 4d does not mention the new RIS-specific knockdown experiment.

Reviewer #2 (Remarks to the Author):

The revised manuscript has addressed my comments. I recommend publication.

Reviewer #3 (Remarks to the Author):

I think the authors have done a good job addressing reviewer comments. I am supportive of publication in the current form.

Reviewer #1 (Remarks to the Author):

After reading the revised manuscript and the responses to all the reviewer comments, my opinion of the paper remains largely unchanged. The identification of the predator cue and the sensory neuron/genetic mechanisms of sensing and reacting to this cue are all very nice and appear robust. The impact of sertraline on avoidance, however, remains less satisfying. In fact, I think it's fair to say that the evidence for sertraline acting specifically on predator avoidance is actually weaker now that two previous conclusions have been reversed (that sertraline does not affect avoidance of other cues, and that *snf-10* mutants retain sertraline modulation.) As an aside, the fact that two repeated experiments failed to replicate raises some concerns about the experiments that were not repeated, but I'm willing to accept that this may be a low probability event and the other conclusions are sound.

I would be more inclined to support publication based on the identification of the predator derived cue and the sensory detection of that cue, if it were not for the fact that the sertraline result, and the implication that this suggests some conserved (or convergent) mechanism for stress/anxiety-driven predator responses appears to remain a cornerstone of the paper. I fear that the publicised narrative will be that worms get stressed by predators and their anxiety can be relieved with antidepressants. While this may be true, the evidence that there is a "conserved or convergent strategy for managing predator threats" is weak. This is a drug that may have all kinds of effects on different neurotransmitter systems. The fact that it (nonspecifically) reduces avoidance to a cue that happens to be secreted by a predator is not great evidence for functional conservation.

If the effects of sertraline and the idea of functional conservation were presented as more preliminary and tentative, I would be more supportive of the paper. Whether this weakens the impact is a question that is probably best dealt with editorially.

We acknowledge the reviewer's concern that our sertraline data is preliminary and we have edited our manuscript to reflect this. However, our claim of a conserved or convergent mechanism is based on our results that *C. elegans* uses multiple redundant circuits and signaling pathways including CNG and TRP channels.

Two more minor comments relating to the revision:

1) On line 195 it is stated that sertraline "affects some, but not all repellent circuits". Given that it affects all three cues tested, it's not clear to me how the "some, but not all" part of this statement is justified.

We find that *C. elegans* responses to copper sulfate is not modified by sertraline treatment, which justifies this statement.

2) The figure legend for Figure 4d does not mention the new RIS-specific knockdown experiment.

We have edited the figure legend for 4d.

Reviewer #2 (Remarks to the Author):

The revised manuscript has addressed my comments. I recommend publication.

Reviewer #3 (Remarks to the Author):

I think the authors have done a good job addressing reviewer comments. I am supportive of publication in the current form.